# Achieving Remission in the Era of Clinical Inertia: What Is Preventing Us from Treating Type 2 Diabetes?

**Austen Suits** [ID]**, Ridhi Gudoor and Jay H. Shubrook \*** [ID]

College of Osteopathic Medicine, Touro University California, Vallejo, CA 94592, USA
\* Correspondence: jshubroo@touro.edu

**Abstract:** Despite evolution in treatment options and improved understanding of pathophysiology, the treatment of type 2 diabetes remains unsatisfactory. Current management guidelines complicated by clinical inertia have resulted in over half of patients failing to meet glycemic targets. Expert consensus has defined a state of diabetes remission whereby treatment can induce sustained normalization of glucose levels. Evidence suggests that metabolic surgery, intensive lifestyle modification, and pharmacologic approaches are each viable options for achieving remission when implemented early in the disease course. The authors review each of these strategies and include practical considerations to aid in the pursuit of remission.

**Keywords:** diabetes; clinical inertia; remission

## 1. Introduction

### 1.1. Current and Future Burden of T2DM

Type 2 diabetes mellitus (T2DM) is a non-communicable pandemic. Diabetes is frequently encountered regardless of specialty, and the disease burden is worsening each and every year. An estimated 37.3 million people (11.3%) in the United States have diabetes, with another 1.5 million cases expected per year [1]. An additional 96 million people (38%) have prediabetes with a high risk of progression to T2DM [1].

Diabetes mellitus treatment cost is staggering, with a majority of expenses attributable to treatment of late-stage complications, including cardiovascular and renal disease. Care for patients diagnosed with diabetes accounts for approximately 25% of every dollar spent on healthcare in the United States, more than any other disease [2].

### 1.2. An Inadequete Implementation of the Standard of Care

There is a plethora of treatment options available for patients with T2DM, yet less than half achieve treatment goals [3]. Current conventional clinical practices have proven largely unsuccessful in attaining adequate glycemic control and maintaining HbA1c under the recommended standard of 7.0% [4]. Importantly, this is not due to a failure of therapeutic innovation. Over the last 20 years, several novel and promising antidiabetes medications and technologies have been developed with consistent improvements in glycemic efficacy. Even metabolic surgical methods have evolved along with intensive lifestyle modification plans.

Despite these encouraging improvements, uncontrolled T2DM continues to have a worsening trend, a troublesome finding given expected outcomes [4,5]. Given that evidence demonstrates that early glycemic control is paramount to avoiding microvascular complications and disease progression, this standard of care is unacceptable.

### 1.3. What Can Be Done?

Evidence supports that when implemented promptly and correctly, a variety of treatment options can result in T2DM remission. The recent ADA consensus report defined

remission as an HbA1c < 6.5% (48 mmol/mol) for at least 3 months without use of glucose-lowering medication [6]. Per this definition, multiple methods have been demonstrated in clinical trials to induce remission. While not every patient with T2DM will achieve remission, this evidence suggests it is a reasonable treatment goal for all patients who are newly diagnosed with diabetes.

In this manuscript, the authors review various treatment methods that have been able to induce T2DM remission and introduce the concept of a first-year plan that emphasizes the importance of early glycemic control and thoughtful selection of an aggressive, individualized treatment plan.

## 2. Keys to Effective Treatment

### 2.1. The Beta Cell

Knowledge of the pathophysiology of T2DM has evolved considerably over the last 20 years. Initially, researchers theorized that glucotoxic mediated apoptosis was predominantly responsible for loss of β-cell function, declining levels of insulin, and progression of the hyperglycemic state [7]. Research now suggests that dedifferentiation is the primary mechanism by which β-cells lose their insulin secreting ability [8]. Furthermore, emerging data demonstrate that a sustained euglycemic state can result in redifferentiation of β-cells, restoring their ability to secrete insulin [9]. However, this responsiveness is disease duration dependent.

The implications of these findings provide new hope for clinicians and patients with T2DM. If treatment is capable of inducing redifferentiation of β-cells, it is plausible that progression of T2DM can be slowed by months to years. Further, such temporization can allow for patients to enter a state of remission, a concept that has now been accepted as a reasonable treatment goal [6].

While the knowledge of T2DM pathophysiology has expanded substantially, differing schools of thought remain on the best way to effectively treat T2DM and lower HbA1c. Multiple studies support each theory, suggesting that remission can be achieved through multiple avenues.

### 2.2. From the Triumvirate to the Egregious Eleven

In 1987, Ralph DeFronzo established three main components responsible for the pathogenesis of T2DM [10]. This theory outlined β-cells, muscle, and the liver as the triumvirate accountable for the development of glucose intolerance and overt T2DM [10]. What started as the triumvirate became the ominous octet, and now has advanced to the egregious eleven [11,12]. Each of these distinct pathological processes has a unique contribution to the disease, and each new physiological finding brings potential for development of novel treatment options.

At the center of these 11+ factors are two main mechanisms: declining tissue sensitivity to insulin followed by the deteriorating ability of β-cells to secrete insulin. Figure 1 outlines how each of the currently known mechanisms contribute to this pathophysiology. While adding to the complexity of the disease, discovery of each new mechanism introduces new treatment targets to address.

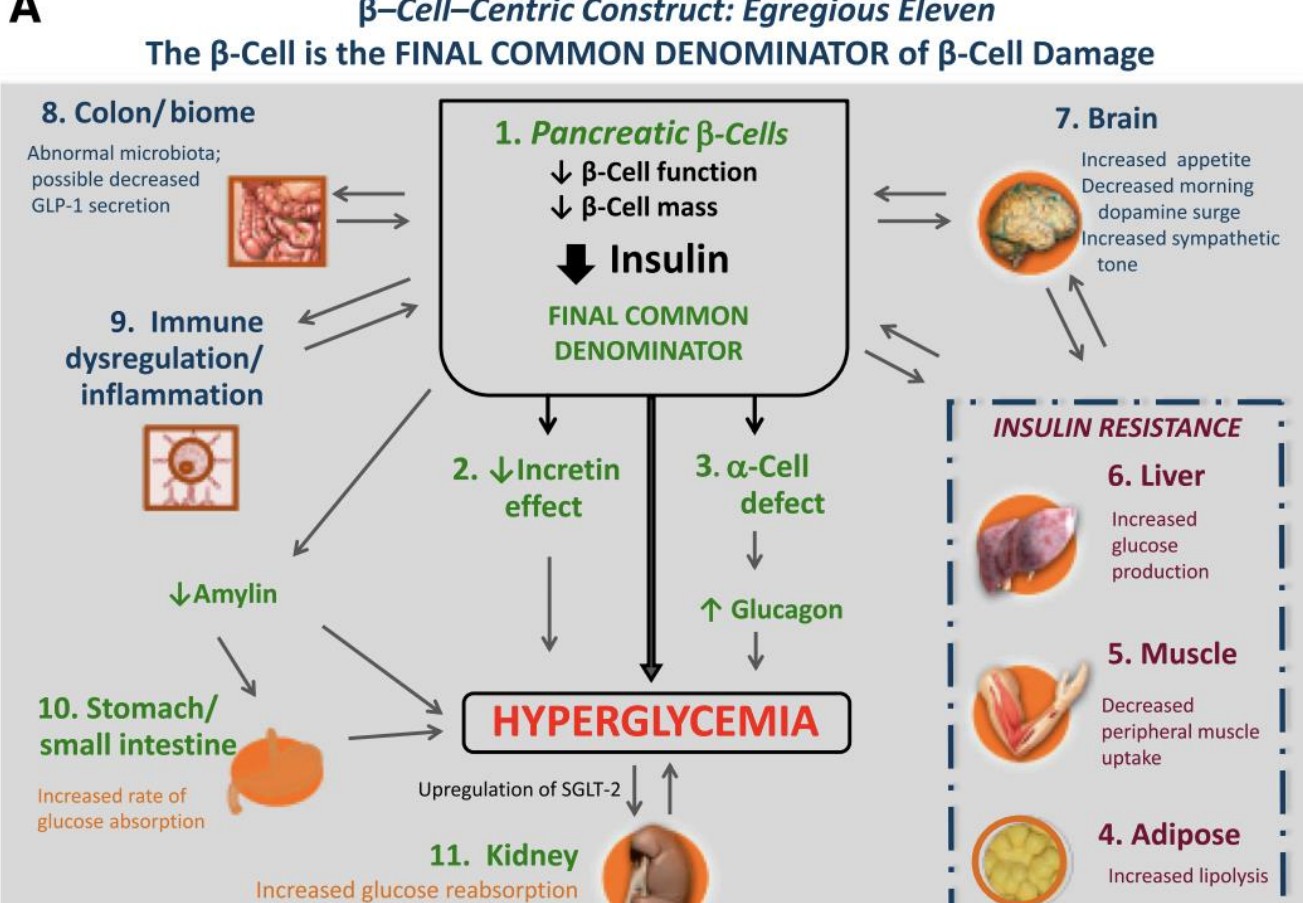

**Figure 1.** Adapted with permission from Schwartz et al. [12]. The egregious eleven. β-cell–centric construct: the egregious eleven. Dysfunction of the β-cells is the final common denominator in DM. Eleven currently known mediating pathways of hyperglycemia are shown. Many of these contribute to β-cell dysfunction (liver, muscle, adipose tissue (shown in red to depict additional association with IR), brain, colon/biome, and immune dysregulation/inflammation (shown in blue)), and others result from β-cell dysfunction through downstream effects (reduced insulin, decreased incretin effect, α-cell defect, stomach/small intestine via reduced amylin, and kidney (shown in green)).

*2.3. The Twin Cycles Hypothesis*

A separate theory established by Ron Taylor suggests that achieving remission requires treatment aimed at halting the "twin cycles" hypothesis [13]. This theory postulates that chronic adipose fat accumulation in the liver will eventually result in fat infiltration into the pancreas (Figure 2). The result is a state of adipose excess in which hepatic sensitivity to insulin declines and β-cells become dysfunctional, causing increasingly elevated plasma glucose and eventually overt T2DM.

Building on evidence of glycemic normalization following metabolic surgery, advocates for this theory posed that inducing a negative caloric balance can decrease intra-organ fat in the liver and pancreas [14,15]. Crucially, they theorized that resolution of this fat accumulation would result in reversal of the twin cycles, thus normalizing function and slowing or even halting the progression of T2DM [14,15].

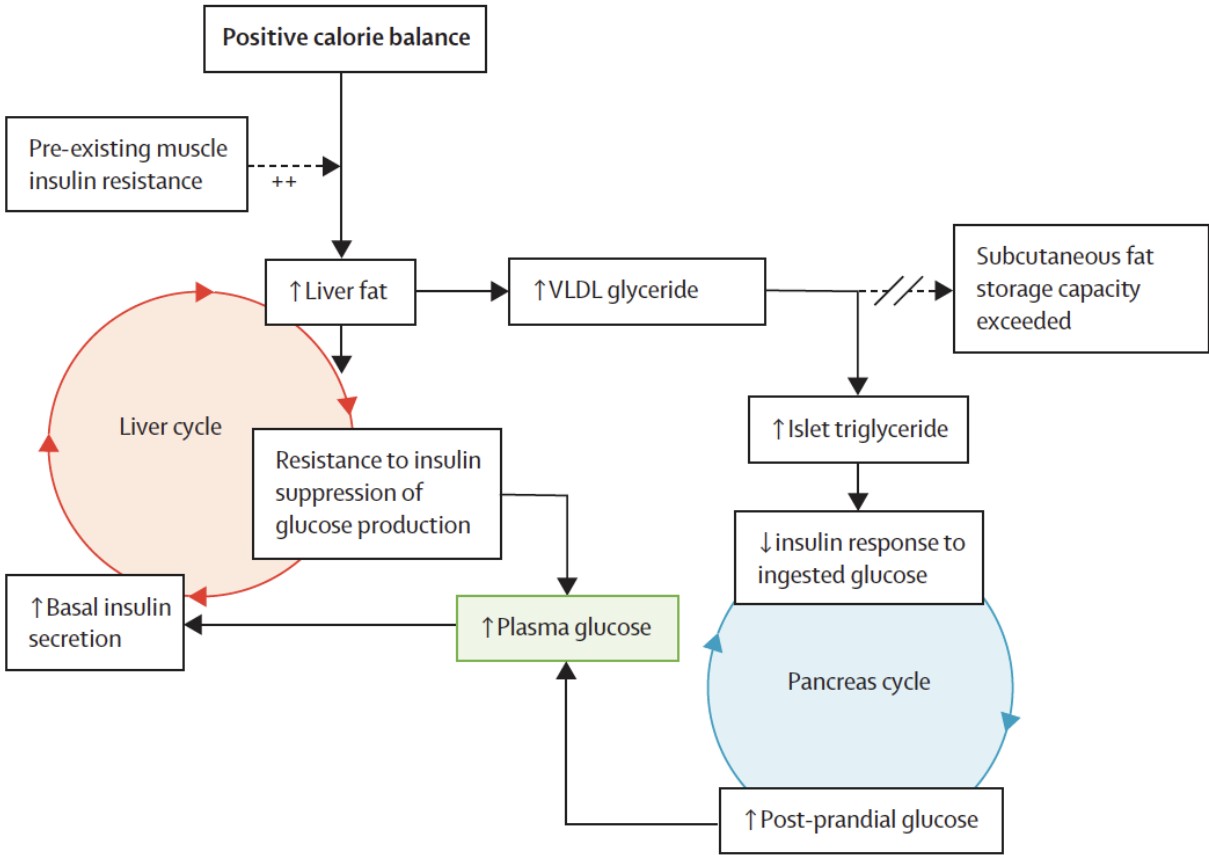

**Figure 2.** Twin cycle hypothesis. During chronic positive calorie balance, the de novo lipogenesis pathway handles carbohydrates which cannot be stored as glycogen, promoting fat accumulation in the liver. Because the process is stimulated by insulin, individuals with a degree of insulin resistance (determined by genetic or lifestyle factors) accumulate liver fat more readily than others due to the higher plasma insulin concentrations. Consequently, the increased liver fat causes resistance to insulin suppression of hepatic glucose production. Over many years, a small increase in fasting plasma glucose leads to increased basal insulin secretion to maintain euglycemia. The consequent hyperinsulinaemia further increases the conversion of excess calories into liver fat. A vicious cycle of hyperinsulinaemia and blunted suppression of hepatic glucose production is thereby established. Fatty liver leads to increased export of VLDLs triglyceride into the circulation (and thus excess ectopic fat in the blood), which increases fat delivery to all tissues, including the pancreatic islets. This process is further stimulated by increased plasma glucose concentrations. Excess fatty acid availability in the pancreatic islets would be expected to impair the acute insulin secretion in response to ingested food, and at a certain level of fatty acid exposure, post-prandial hyperglycaemia supervenes. The hyperglycaemia further increases insulin secretion rates, with consequent increase in hepatic lipogenesis, spinning the liver cycle faster and driving on the pancreas cycle. Eventually the fatty acid and glucose inhibitory effects on the islets reach a trigger level, leading to β-cell failure and a fairly sudden onset of clinical diabetes. (Reprinted with permission from Taylor et al. [13]).

*2.4. Does Superiority Matter?*

Often lost in the debate over the pathophysiology of T2DM is the primary goal of the patient and their physician: effective treatment and prevention of complications. Does the route by which this is achieved matter? The superiority of one theory to another becomes trivial when treatment outcomes continue to decline. Individualized treatment in a primary care setting is necessary to address a variety of patients each with unique circumstances. While one patient may find success with intensive lifestyle modification, another may prefer a combination of medication or surgery. The importance of the above theories is that they provide multiple pathways to achieving diabetes remission. Thus, clinicians should

focus on utilizing the treatment option best suited to each patient with an emphasis on early implementation.

### 3. The Current State of Care: Indications for Urgency

*3.1. Metformin and Current Guidelines*

Since metformin was first introduced in the United States in 1995, it has become the foundational first-line treatment for T2DM. Three decades years later, it remains the most commonly prescribed medication for the initial management of the disease [16]. The vast majority of clinicians utilize a step-up approach to T2DM care. This model of care begins with metformin and culminates with basal/bolus insulin, adding therapy each time glucose is found to be persistently elevated. Further, a variety of new antidiabetes medications have been developed over the past two decades, adding layers of complexity when selecting the proper therapeutic choice for patients. The flaw in this system is that treatment is intensified only when the previous treatment fails. With each treatment failure the disease progresses as clinicians attempt a nearly impossible race to catch up. Each delay in intensification results in significantly increased risk of MI, HF, stroke, and composite CVE [17].

*3.2. Therapeutic Inertia*

Evidence of Therapeutic Inertia in Diabetes Care

Therapeutic inertia is defined as the lack of timely adjustment to a patient's treatment regimen when their treatment targets are not met [18]. This is not simply clinician failure. It is a multifactorial issue affected by a variety of stakeholders, including patients, providers, and health systems [19]. A U.K. Clinical Practice Research Datalink retrospective study indicated that patients with an HbA1c > 7.0% waited an average of 2.9 years before treatment was intensified [20]. Perhaps more concerning, another study revealed that only 56% of primary care providers considered escalating therapy when their patients did not meet glycemic targets [21]. In contrast, ADA recommendations suggest intensifying treatment within 3–6 months if targets are not met [22].

*3.3. The Burden of Inertia*

Failure to intensify treatment comes at a high cost not only to patients, but also to healthcare systems. In a US study evaluating the burdens of therapeutic inertia in T2DM, Ali et al. found that delaying intensification of therapy by 1 year was associated with 13,390 life years lost and increased costs of 7.3 billion USD [23]. Lindvig et al. suggest that early and intensive glycemic control minimizes therapeutic inertia, thereby resulting in substantial monetary savings and decreased economic disease burden [24]. These findings suggest that clinical inertia is detrimental both from a clinical and economic perspective. It also demonstrates that the urgency with which treatment is implemented and progressed can make a difference both clinically and economically.

*3.4. Overcoming Therapeutic Inertia*

In response to the unsettling prevalence of therapeutic inertia, the American Diabetes Association launched the 3-year initiative "Overcoming Therapeutic Inertia" in 2020 [18]. As part of the OTI initiative, researchers investigated the best strategies for overcoming therapeutic inertia. Figure 3 outlines a framework for the best clinical practices discovered through this initiative. Implementation of these tactics can allow for better outcomes and remission when integrated with effective treatment applied early in the disease course.

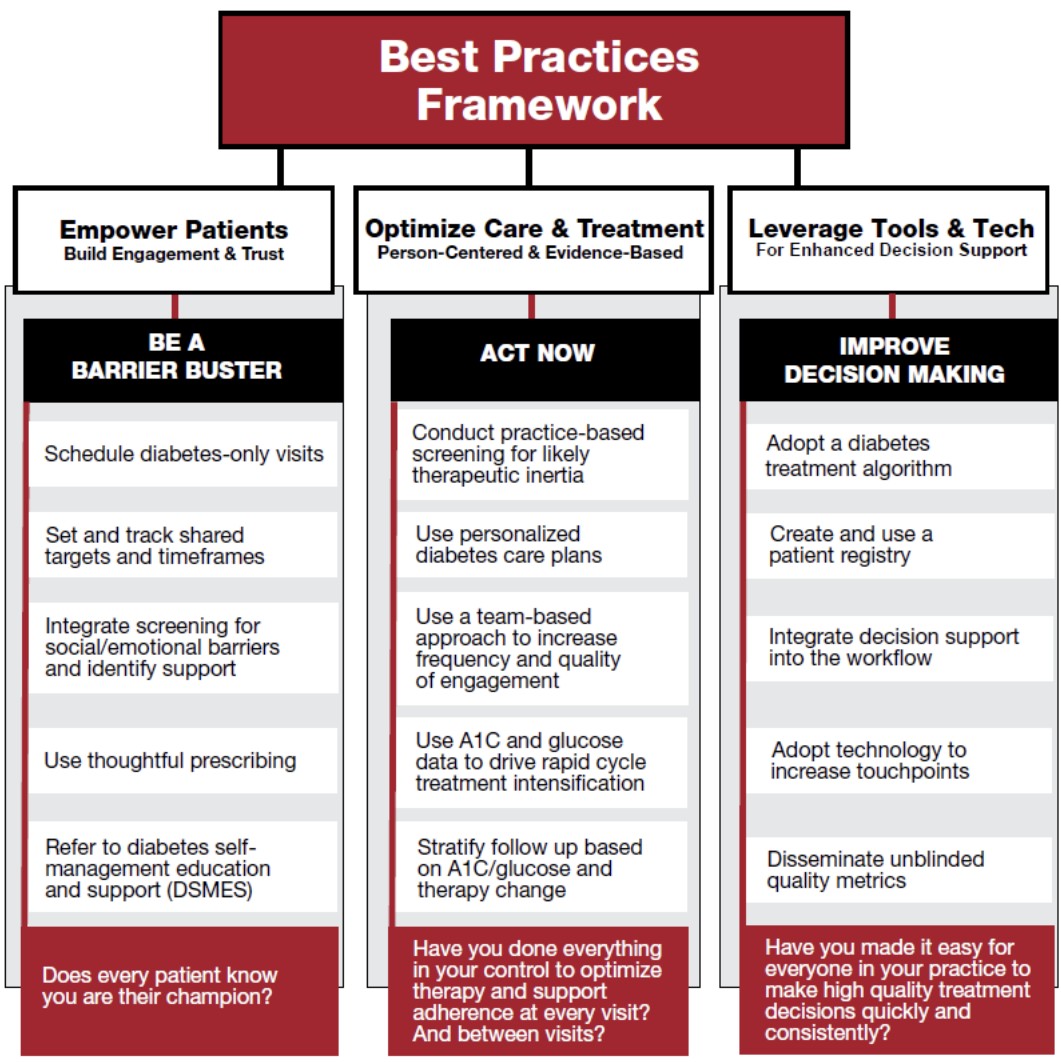

**Figure 3.** OTI initiative best clinical practices. Methods for avoiding clinical inertia based on the ADA's OTI Initiative. These action-based recommendations are evidence- and consensus-based on real practice experiences. Reprinted with permission from Gabbay et al. [18].

*3.5. Future Considerations*

The ADA's initiative has successfully created and promoted important resources to assist clinicians in the proactive treatment of T2DM. While these steps are undeniably important, future consideration is warranted for the adoption of early, intensive methods as first line treatment for T2DM. Multiple intensive treatment methods have demonstrated to be both safe and effective in the treatment of T2DM, and all induce remission at a significantly higher rate than traditional metformin step-up therapy [25–28]. By implementing these methods early, the prevalence of therapeutic inertia may decrease as patients have a better opportunity to achieve initial glycemic control and T2DM remission.

**4. Effective Treatment Options**

Here, the authors review a variety of intensive treatment options, all of which have been shown to induce remission in patients with T2DM when utilized correctly. This review is not intended to substantiate one method over another, but rather explore the variety of tools a provider has to help a patient with T2DM achieve glycemic control. Most importantly, it is meant to emphasize the common features: prompt implementation, continual assessment, and front-loaded intensity of care.

### 4.1. Therapeutic Lifestyle Change
Evidence

Therapeutic lifestyle modification will always be a mainstay in the treatment of T2DM throughout the disease progression. Strong evidence supports that weight loss decreases liver and pancreatic adiposity, thus improving insulin sensitivity [15,29]. Downstream effects include decreased lipotoxicity and improved β-cell function, resulting in improved clinical measures, such as HbA1c [15]. The Diabetes Remission Clinical Trial (DiRECT) demonstrated a viable option for achieving remission in a primary care setting with intensive lifestyle modification alone [26]. Based on the success and lessons gained from the previous Counterpoint and Counterbalance studies, Lean et al. developed a practical and durable dietary modification plan to achieve T2DM remission [30,31].

Patients completed 3–5 months of total diet replacement, followed by 6–8 weeks of stepped food reintroduction and structured support for weight loss maintenance. Results at 12 months demonstrated encouraging findings, with remission in 46% of participants, mean weight loss of 10.0 kg, and mean HbA1c decrease of 0.9% [26]. Furthermore, these results persisted at the 24-month mark with over 1/3 of participants maintaining remission [32].

The results from DiRECT provide one path to remission and give merit to the theory that patients can achieve diabetes remission with intensive diet modification alone.

Importantly, being prescribed fewer antidiabetes medications was the strongest baseline predictor in DiRECT, with patients having a greater likelihood of remission prior to introduction of first- or second-line oral agents [33]. These findings suggest that urgent, early implementation of an intensive weight-management program is imperative for success.

### 4.2. Metabolic Surgery
Evidence

The most well documented treatment option for achieving T2DM remission is metabolic surgery. It follows a similar concept as intensive diet management, hypothesizing that weight loss and negative caloric balance achieved through surgery and thereafter can produce improved β-cell function and remission. Multiple surgical options exist, including sleeve gastrectomy, Roux-en-Y gastric bypass, and laparoscopic adjustable gastric banding.

The STAMPEDE trial first demonstrated superior efficacy of metabolic surgery compared to traditional medical treatment in T2DM [25]. Numerous other trials have followed with similar results. Figure 4 summarizes the results of twelve randomized controlled trials, with only one failing to demonstrate statistical significance of metabolic surgery. Recently, ARMMS-T2D study investigators published promising 3-year outcomes from a combination of four major randomized controlled trials. Results indicated that T2DM remission was achieved in 37.5% of patients following surgery compared to 2.6% of medical/lifestyle intervention patients, with superior reductions in HbA1c, FPG, and BMI [34]. These findings are durable, with a meta-analysis of remission from T2DM 5–15 years post-surgery exhibiting an approximate six-fold increase in remission compared to conventional therapy [35].

Similar to findings in lifestyle modification studies, baseline characteristics that positively predict remission following surgery include shorter duration of T2DM and better β-cell function [35]. This once again exemplifies the importance of avoiding clinical inertia and promptly implementing care in order to achieve the most favorable outcome possible.

Metabolic surgery for type 2 diabetes mellitus: Randomized controlled clinical trials.

| Study | Pts with BMI < 35 kg/m³ | Study design | No. pts | Follow-up (mo) | Remission Criteria | Remission ª or change in HbA1c (%) | P value |
|-------|-------------------------|--------------|---------|----------------|--------------------|-----------------------------------|---------|
| Dixon | 22% | LAGB vs control | 60 | 24 | HbA1c < 6.2% | 73 vs 13 | <0.001 |
| Schauer | 36% | RYGB vs SG vs control | 150 | 60 | HbA1c < 6.0% | 22 vs 15 vs 0 | <0.05 |
| Mingrone | 0% | RYGB vs BPD vs control | 60 | 60 | HbA1c < 6.5% | 42 vs 68 vs 0 | 0.003 |
| Ikramuddin | 59% | RYGB vs control | 120 | 24 | HbA1c < 6.0% | 44 vs 9 | <0.001 |
| Liang | 100% | RYGB vs control | 101 | 12 | HbA1c < 6.5% | 90 vs 0 vs 0 ᵇ | <0.0001 |
| Halperin | 34% | RYGB vs control | 38 | 12 | HbA1c < 6.5% | 58 vs 16 | 0.03 |
| Courcoulas | 43% | RYGB vs LAGB vs control | 69 | 36 | HbA1c < 6.5% | 40 vs 29 vs 0 | 0.004 |
| Wentworth | 100% | LAGB vs control | 51 | 24 | FBG < 7.0 mmol/L | 52 vs 8 | 0.001 |
| Parikh | 100% | RYGB/LAGB/SG vs control | 57 | 6 | HbA1c < 6.5% | 65 vs 0 | 0.0001 |
| Ding | 34% | LAGB vs control | 45 | 12 | HbA1c < 6.5% | 33 vs 23 ᶜ | 0.46 |
| Cummings | 25% | RYGB vs control | 43 | 12 | HbA1c < 6.0% | 60 vs 5.9 | 0.002 |
| Shah | 85 | RYGB vs control | 80 | 24 | HbA1c < 6.5% | 60 vs 2.5 | <0.001 |

**Figure 4.** Metabolic surgery randomized clinical trials. Remission criteria: ª Remission was primary or secondary end point; HbA1c value without diabetes medications, unless otherwise specific. ᵇ Remission was not precisely defined; HbA1c < 6.5% by extrapolation. ᶜ Intermittent diabetes medications. BMI = body mass index; BPD = biliopancreatic diversion; FBG = fasting blood glucose; HbA1c = glycated hemoglobin; LAGB = laparoscopic adjustable gastric band; RYGB = Roux-en-Y gastric bypass; SG = sleeve gastrectomy.

### *4.3. Pharmacologic Approaches*

The current "treat to failure" model of T2DM care is indeed failing. Initiating metformin and adding medication after failing to meet targets leaves room for clinical inertia and has limited value in achieving remission. However, early intensive pharmacologic approaches with combination therapy and insulin deserve consideration for patients preferring medication over lifestyle modification or surgery.

### *4.4. Combination Therapy*

To test the hypothesis that targeting the ominous octet could provide durable reduction of HbA1c, DeFronzo and colleagues designed a triple combination therapy for drug naïve patients with T2DM. The study, EDICIT, demonstrated that initial treatment with a combination of metformin/pioglitazone/exenatide provided a significantly greater reduction in HbA1c compared to conventional step-up therapy with metformin, followed by sequential addition of a sulfonylurea and glargine insulin [27]. Importantly, 61% of patients receiving triple therapy had HbA1c reduced to within normal range (<6.0%) and decrease in HbA1c was sustained over a two-year period. At three-year follow-up, 67% of patients receiving triple therapy had maintained HbA1c < 6.5% [36].

The importance of the EDICT study is twofold. First, it demonstrates that an intensive initial treatment of T2DM with combination therapy is both safe and therapeutically superior to traditional step-up therapy. Patients on triple therapy had a significantly lower frequency of hypoglycemic events, and the most common adverse effect was mild nausea [36]. Secondly, it demonstrates a viable path to avoiding several of the components of therapeutic inertia. By initiating therapy with a combination of medication, it minimizes the necessary steps of escalation required by traditional therapy. This provides an accelerated path to glycemic control, which by extension should result in better patient outcomes.

The EDICT trial provides optimism about the value of initial combination therapy, while the evolution of antidiabetes medications offers the potential for new combinations which may improve outcomes. The emergence of semaglutide suggests a more potent GLP-1RA as demonstrated in the SUSTAIN and STEP trials [37–39]. The new dual-incretin tirzepatide has also been demonstrated to provide significant HbA1c reductions and substantial weight loss as outlined in the SURPASS and SURMOUNT trials [40,41]. SGLT2i address the pathophysiological mechanism of the kidney in T2DM, the only piece of the ominous octet not addressed in EDICT [42]. The landscape of pharmacologic options for T2DM treatment is constantly expanding, and research addressing new combination options to induce remission must augment these innovations.

### 4.5. Intensive Insulin Therapy (IIT)

Insulin therapy is appropriate at any point in T2DM progression but is rarely used prior to failure of oral therapies and lifestyle modification. Multiple clinical trials have demonstrated the ability of early IIT to induce T2DM remission, improve β-cell function, and maintain glycemic control [43,44]. Furthermore, a 2013 meta-analysis revealed that IIT is associated with an approximate 46% rate of drug-free remission at 12 months post-treatment [28]. While a common concern of insulin use is safety, assessment of adverse effects has demonstrated no significant difference in hypoglycemic risk, anthropometry, or quality of life between treatment with IIT versus metformin [45].

While most short course IIT clinical trials (4–12 weeks) have involved intravenous insulin in an inpatient setting, INSPIRE trial data provide evidence that IIT is feasible in an outpatient setting, and results in dramatic glucose reduction without weight gain or hypoglycemia [46].These findings suggest IIT is a viable option for first-line treatment to induce T2DM remission.

### 4.6. Remission with Maintenance Therapy

Despite the significant reductions in HbA1c demonstrated in EDICT and IIT trials, uncertainty remains about the durability of remission following the removal of medications. The LIBRA trial demonstrated that following cessation of treatment with liraglutide, the beneficial effect of treatment was lost [47]. If remission cannot be maintained without the use of antidiabetes medications, is it still worth the endeavor?

Recently, experts who authored the consensus report on remission suggested potential for a second definition of remission with ongoing drug therapy. Importantly, experts stated that striving for remission should not prevent use of medications that provide substantial benefit. Given that glycemic control per ADA recommendations can exist at an HbA1c of up to 8%, value exists for a new definition in which a patient maintains an HbA1c below the diagnostic threshold for T2DM (<6.5%). Prior studies have demonstrated deleterious microvascular effects beginning in the prediabetes HbA1c range [48,49]. Thus, the discussion of distinction of remission with or without medication should not surpass the importance of addressing T2DM pathophysiology to achieve a euglycemic state, even if continued medication use is required.

## 5. Important Considerations When Selecting Treatment

### 5.1. Importance of Early, Intensive Care

Irrespective of the modality of treatment chosen, it is imperative for clinicians to urgently institute effective care and avoid clinical inertia. The importance of early glycemic control in T2DM has been well established and is critical to preventing diabetes-related complications [50]. Given the progressive nature of insulin resistance and β-cell dysfunction, by the time a patient is diagnosed with overt T2DM, pathology has been escalating for years.

It is estimated that up to 80% of β-cell function is lost by the time of diabetes diagnosis, imploring clinicians to select treatments shown to improve and preserve β-cell function [51]. Each of the aforementioned treatment methods has been shown to fulfil these criteria, while metformin has not. Each method also predicts improved chance of remission if applied early in the disease course. To give patients an optimal chance at remission, effective treatment must be part of the initial plan, not a reinforcement following treatment failure.

### 5.2. Collaborative First-Year Plan

The continuously increasing rate of β-cell dysfunction makes the first year of diagnosis extremely important in the T2DM disease course. While each patient with T2DM requires a unique treatment plan suited to their specific disposition, all patients deserve a plan that empowers them to achieve early glycemic control. Utilizing shared decision making, setting and tracking realistic goals, and promptly escalating care when indicated, can set patients up for success. Figure 5 illustrates a potential flow chart to assist patients and providers in selecting a plan that offers the best chance for remission.

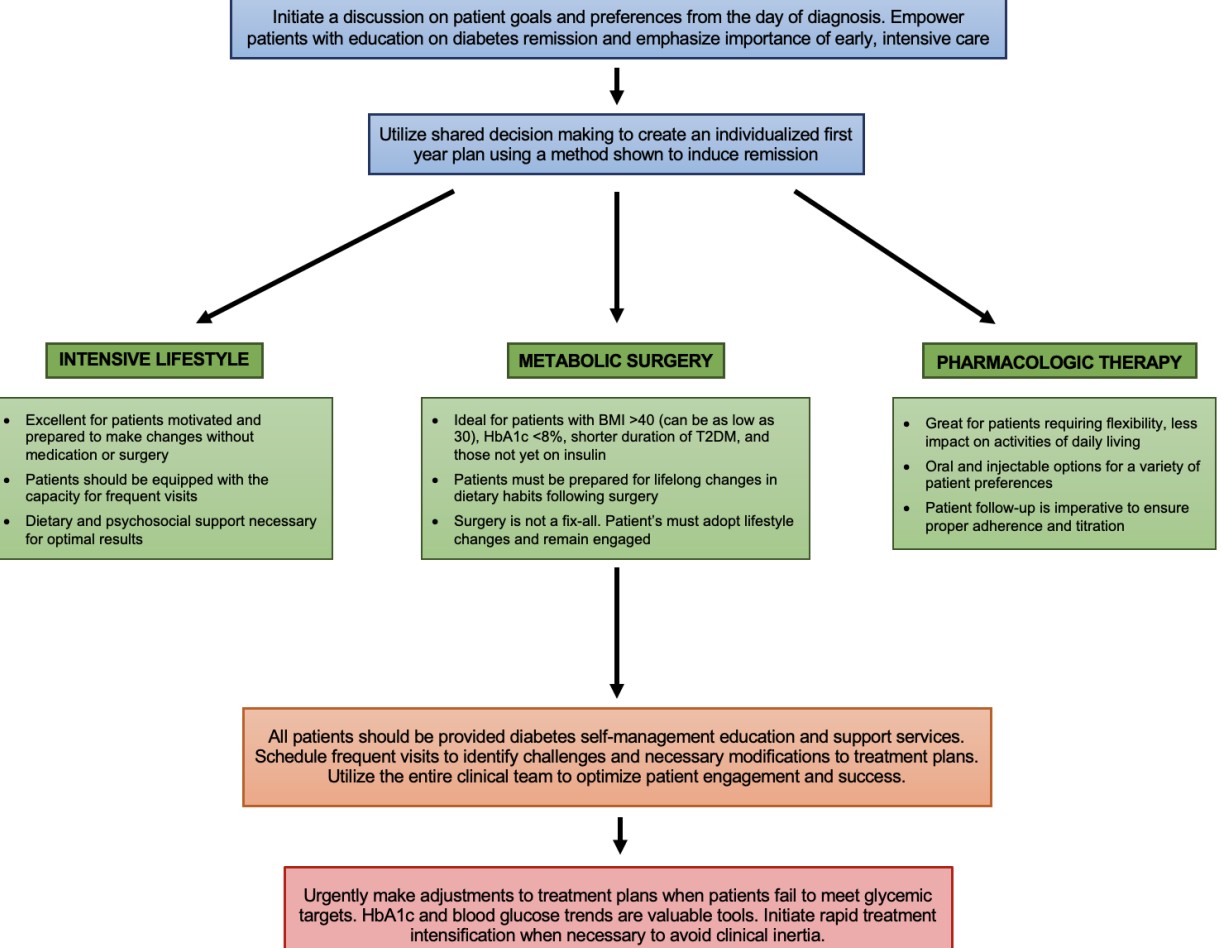

**Figure 5.** Creating a first-year plan to achieve remission. A flowchart describing methods shown to induce remission and practical application of a first-year plan. Intensive lifestyle refers to DiRECT protocol and similar intensive lifestyle modification programs. Metabolic surgery refers methods including Roux-en-Y gastric bypass, sleeve gastrectomy, and laparoscopic adjustable gastric banding. Pharmacologic therapy refers to the combination therapy or intensive insulin therapy.

## 5.3. Choosing a Treatment

Intensive Lifestyle Modification

This method of achieving glycemic remission could be particularly helpful in younger patients diagnosed with T2DM, who tend to develop increased obesity and have a more rapid disease progression [52]. Additionally, patients with an aversion to medication or surgery are likely to find this treatment option more appealing, albeit challenging. Patients who do not prefer a lifestyle alone approach to treatment should still develop a structured lifestyle modification plan in conjunction with medication or surgery.

Given the rigorous and potentially overwhelming nature of intensive lifestyle modification, it is important for clinicians to provide proper education prior to treatment initiation. Patients will need a team-based approach, including advanced nutritional and psychosocial support. Consideration should be given to family and friends who will see changes in eating patterns. Patients who fail to achieve or maintain remission should be promptly initiated on additional therapy as to avoid worsening glycemic control and clinical inertia.

## 5.4. Metabolic Surgery

Currently, the ADA recommends metabolic surgery for patients with T2DM and a $BMI > 40 \, kg/m^2$ ($BMI > 37.5 \, kg/m^2$ in Asian Americans) and in patients with a $BMI > 35 \, kg/m^2$ who cannot achieve durable weight loss and improvement in comorbidities with nonsurgical methods [53]. However, a recent study demonstrated that Roux-en-Y gastric bypass surgery in patients with a BMI of 25–32.5 resulted in T2DM remission in 60.6% of patients, indicating potential value for larger population of patients with T2DM regardless of BMI status [54]. Furthermore, independent predictors of long-term remission indicate metabolic surgery is most effective for those with short duration of T2DM, HbA1c < 8%, and those using fewer antidiabetes medications and not yet on insulin [55].

Remission rates appear to be consistently higher in patients who undergo surgical intervention, however there are known medical risks to metabolic surgery. While rare, there is an approximate 0.3% 30-day mortality rate and 4.3% incidence of major postoperative adverse events following surgery [56]. Patients may encounter nutritional deficiencies which can lead to anemia or decreased bone mineral density; however, these can be corrected with proper assessment and supplementation. Thus, a collaborative discussion with potential candidates is important to ensure surgery is the correct option versus lifestyle modification or medication. Additionally, patients should be made aware of potential future need for lifestyle modification or medication use post-surgery to maintain weight loss and glycemic control.

Medication

Convenience of care is an important consideration when determining the ideal treatment for a patient. For many with T2DM, medication is a convenient alternative to lifestyle modification or metabolic surgery. While still intensive, medication combination therapy and IIT can provide patients with the flexibility of treatment at home, fewer dietary constraints, and less disruption to activities of daily living. Patients should be thoroughly screened for contraindications to combination therapy or IIT before starting treatment. Additionally, patients will need continuing education and support to ensure they are titrating medications properly and to decrease risk of adverse events.

## 6. Room for Improvement: Current Limitations and Future Considerations

While this review intends to provoke inspiration for a new approach to T2DM treatment focused on early implementation and remission, practical limitations must be considered. Currently, the literature available contains relatively small numbers of patients with short duration of follow-up. Large scale, multicenter studies over an extended time period are warranted to assess the feasibility and durability of the treatment methods described. Relapse rates must be considered, and it is certainly possible that providers and patients will require multiple methods and strategies in the quest for durable remission. Further, we

do not know yet if diabetes remission also means reduction in other outcomes, especially CV outcomes. A recent article highlighted the value of making weight loss the focus of type diabetes treatment. The approach is consistent with the findings in this manuscript.

Although many patients may find significant therapeutic value in the methods described in this review, certain populations may not be suitable for the intensity of these interventions. Special consideration must be given to patients with multiple comorbidities, frailty, and the elderly to ensure treatment is appropriate. A thorough discussion of risks and benefits of treatment is essential to ensuring positive outcomes for all patients with T2DM.

The feasibility of treatment implementation in a primary care setting must also be a priority consideration. Many providers, practices, and healthcare systems are not currently equipped to provide a patient with T2DM the comprehensive options of intensive lifestyle modification, metabolic surgery, and initial pharmacologic combination therapy. We have unequal distribution of necessary care teams and resources. This can make community implementation of study trial interventions more challenging. Increasing the prevalence of remission will require dynamic adaptation from each of these stakeholders and an overhaul of currently utilized approaches.

## 7. Conclusions

We can and must do better in treating T2DM. Inadequate first-line treatment and therapeutic inertia has resulted in worsening glycemic trends, increasing disease burden, and unacceptable outcomes. Despite this, current understanding of T2DM pathophysiology and evolution of treatment methods has made remission achievable. Intensive lifestyle modification, metabolic surgery, and pharmacologic therapies are all viable options to help your patient reach this realistic goal. Each of these treatments requires early implementation, continuous assessment, and timely adjustment. The importance is not in the route by which remission is achieved, but in selecting a treatment that gives your patient a chance for remission from the first day of diagnosis.

**Author Contributions:** A.S. contributed to conception, design, drafting and critical revision. R.G. contributed to design and critical revision. J.H.S. contributed to conception, design, drafting and critical revision. All authors have read and agreed to the published version of the manuscript.

**Funding:** This research received no external funding.

**Institutional Review Board Statement:** Not applicable.

**Informed Consent Statement:** Not applicable.

**Data Availability Statement:** The data presented in this study are available on request from the corresponding author.

**Conflicts of Interest:** AS has no relevant conflicts to disclose. RG has no relevant conflicts to disclose. JHS has served as an advisor to Abbott, Astra Zeneca, Bayer, Eli Lilly, Nevro, and NovoNordisk.

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
