# Peer review of "Achieving Remission in the Era of Clinical Inertia: What Is Preventing Us from Treating Type 2 Diabetes?"

_diabetology, doi:10.3390/diabetology4010011_

Round 1

Reviewer 1 Report

Amazing though provoking paper.  One of the most comprehensive I have read.  A definite call to arms.  So concise.  I like how it mentions remission with sustained pharmacology because if ongoing medication is required for remission then so be it.  

Author Response

Response to Reviewers:

Reviewer 1:

Comments and Suggestions for Authors

Amazing though provoking paper.  One of the most comprehensive I have read.  A definite call to arms.  So concise.  I like how it mentions remission with sustained pharmacology because if ongoing medication is required for remission, then so be it.  

RESPONSE: Thank you for your thoughtful review. We were hoping it would be seen as thought-provoking and a call to arms!

Reviewer 2 Report

Well written review, clear English and clinically relevant topic. In general, I agree with the authors on their views and their wish to highlight the importance of early intervention for newly diagnosed diabetes with the aim of remission of the disease rather than just glycaemic control as the main goal of therapy. 

The review however, seems to be very enthusiastic about this goal although the literature available included only small studies with small numbers of patients and short duration of follow up.  In addition, authors did not mention any CV outcomes, if any, in these studies.   Whether the remission of diabetes also reduced the CV risk down to non-diabetic persons, is not reported or may not be known yet.

The sections on intensive oral or insulin therapy need to expand on whether remission was maintained after withdrawal of therapy or remission was maintained on therapy. In this case, what will be the difference between remission on therapy and tight glycaemic control on therapy? Are they the same thing? Needs discussion please.

I  also suggest that authors include sections on:

1. Limitation of the review just before the conclusion section.  Limitations should address the paucity of a long term evidence on larger study sample, relapse rates, tight glycaemic control or remission on therapy (if no data available), some groups of patients will not be suitable for this kind of intervention such as elderly, multiple morbidities or frailty,..etc  

2. Another section on future perspectives for authors' recommendations on future directions of research. 

Author Response

Reviewer 2:

Comments and Suggestions for Authors

Well written review, clear English and clinically relevant topic. In general, I agree with the authors on their views and their wish to highlight the importance of early intervention for newly diagnosed diabetes with the aim of remission of the disease rather than just glycaemic control as the main goal of therapy. 

The review, however, seems to be very enthusiastic about this goal although the literature available included only small studies with small numbers of patients and a short duration of follow up.  In addition, authors did not mention any CV outcomes, if any, in these studies.   Whether the remission of diabetes also reduced the CV risk down to non-diabetic persons, is not reported or may not be known yet.

The sections on intensive oral or insulin therapy need to expand on whether remission was maintained after withdrawal of therapy or remission was maintained on therapy. In this case, what will be the difference between remission on therapy and tight glycaemic control on therapy? Are they the same thing? Needs discussion please.

RESPONSE: Thank you for your thoughtful review. We agree that the supporting evidence is starting to grow- we want to highlight this to provoke more focus in this area. The concerns about study size/duration of follow-up have now been addressed in the revised manuscript as suggested.

We will comment on CVOTs but we feel that this is outside the scope of this article as a focus. This information is well described in other literature. We find the positive CVOTs to provide additional support and will call that out but refer to other literature that has already described these benefits. We also will acknowledge that we do not know about CV benefits of these remission-focused treatments yet.

We have expanded the benefits of intensive insulin therapy on remission and explained the durability of remission versus remission on therapy. This can be found section Remission with Maintenance Therapy. Tight glycemic control today in everyday practice is mediocre at best so we believe we need to do better than today’s tight glycemic control.

I also suggest that authors include sections on:

  1. Limitation of the review just before the conclusion section.  Limitations should address the paucity of a long term evidence on larger study sample, relapse rates, tight glycaemic control or remission on therapy (if no data available), some groups of patients will not be suitable for this kind of intervention such as elderly, multiple morbidities or frailty,..etc  

RESPONSE: Excellent suggestion—this has been added.

  1. Another section on future perspectives for authors' recommendations on future directions of research. 

RESPONSE: Excellent suggestion—this has been added in the revision in the “Room for Improvement: Limitations and Future Considerations” section. We will also highlight that we expect to get more evidence as we see more study results with medications like the twincretins.

Reviewer 3 Report

1. Only Type 2 Diabetes Mellitus is reversible or can go into remission. Please be consistent in using the terms "Diabetes", "Diabetes mellitus" and T2DM. I would suggest use the acronym T2DM to maintain consistency across the article. 

2. In your 2nd Paragraph, please explain what does " Troublesome finding given expected outcomes" mean ? what are those expected outcomes ? 

3. Please mention below the figure 1, that it has been adapted from Schwartz et al, {reference 12}

4. please use the word 'emphasis' instead of 'emphasize' in paragraph 'does superiority matter?'

5. "The Current state of care: indications for urgency". What does this mean ? 

6. in the paragraph "burden of inertia" please use "urgency with which treatment is implemented"

7. In paragraph future considerations. consider using 'may' instead of 'will', since you are assuming that. Will is a strong word. 

8. Consider using variety of 'tools' a provider has, instead of 'options' or use 'treatment options'

9. you have written Liver and pancreatic adipose, whereas it should be either 'adiposity' or adipose tissue. 

10. in the paragraph therapeutic lifestyle change, in evidence section, The references 30,31 next to counterpoint and counterbalance studies, taylor et al, dont match. please correct that paragraph. 

11. you mentioned "urgent early implementation of an intensive weight management program is imperative'. Did you realize this was part of a study, and not every physician can have access to a weight management program. is there a plan for giving access to patients this weight program. Nor every patient would like complete meal replacement program. 

12. instead of using 'target are not met too often', consider using 'after failing to reach targets'

13. in IIT, yes, early insulin protects pancreatic b cell from damage, but what about the weight gain with insulin. what is the plan to get people off insulin and make them lose weight ? Many patients are not willing to go on insulin. it doesnt mention how long they will be on insulin. this approach of insulin therapy is rarely used these days for diabetes remission, since Bgs are good on insulin. 

14. what is the reference for 80% of b cell function lost. that seems a high number.

14. The sub paragraph ' therapeutic lifestlye change' has been used twice in the article. please consider renaming it. 

15. There is talk about triple combo therapy with metformin/pioglit/exenatide, which is almost 10 years older therapy and study. The newer 2nd gen agents like GLP1, GLP-GIP combo, SGLT2, metformin all have shown to have varied degrees of weight loss. There is not much info or studies on these patients. ozempic or mounjaro{step or sustain or surpass trials]  trials showing dm remission have not been mentioned. DM remission paper cannot be complete without mention of these newer gen medications which have changed the landscape of DM therapy. 

16. Consider incorporating this paper in your article. 

Lingvay I, Sumithran P, Cohen RV, le Roux CW. Obesity management as a primary treatment goal for type 2 diabetes: time to reframe the conversation. Lancet. 2022 Jan 22;399(10322):394-405. doi: 10.1016/S0140-6736(21)01919-X. Epub 2021 Sep 30. Erratum in: Lancet. 2022 Jan 22;399(10322):358. PMID: 34600604. 

17. This article in its current form seems very superficial, and doesnt dive deep. There are many good aspects of the paper, and the topic chosen is important and relevant. 

Hope you can make the paper even better. 

Author Response

Reviewer 3:

Comments and Suggestions for Authors

  1. Only Type 2 Diabetes Mellitus is reversible or can go into remission. Please be consistent in using the terms "Diabetes", "Diabetes mellitus" and T2DM. I would suggest use the acronym T2DM to maintain consistency across the article. 

RESPONSE: Excellent point—this has been clarified.

  1. In your 2nd Paragraph, please explain what does " Troublesome finding given expected outcomes" mean ? what are those expected outcomes ? 

RESPONSE: One would expect that the improvement of treatment options, technology, and understanding of pathophysiology would result in improved outcomes.

  1. Please mention below the figure 1, that it has been adapted from Schwartz et al, {reference 12}

RESPONSE:  Thank you, we will add this language and permission has been granted.

  1. please use the word 'emphasis' instead of 'emphasize' in paragraph 'does superiority matter?'

RESPONSE: Thank you. This has been changed.

  1. "The Current state of care: indications for urgency". What does this mean ? 

RESPONSE: This is in reference to the conventional application of the guidelines for treatment of T2DM and why data suggests a more urgent approach to glycemic control is warranted

  1. in the paragraph "burden of inertia" please use "urgency with which treatment is implemented"

RESPONSE: Thank you. This has been changed.

  1. In paragraph future considerations. consider using 'may' instead of 'will', since you are assuming that. Will is a strong word. 

RESPONSE: Thank you. This has been changed.

  1. Consider using variety of 'tools' a provider has, instead of 'options' or use 'treatment options'

RESPONSE: Thank you. This has been changed.

  1. you have written Liver and pancreatic adipose, whereas it should be either 'adiposity' or adipose tissue. 

RESPONSE: Thank you. This has been changed.

  1. in the paragraph therapeutic lifestyle change, in evidence section, The references 30,31 next to counterpoint and counterbalance studies, Taylor et al, dont match. please correct that paragraph. 

RESPONSE: Thank you. This has been changed.

  1. you mentioned "urgent early implementation of an intensive weight management program is imperative'. Did you realize this was part of a study, and not every physician can have access to a weight management program? is there a plan for giving access to patients this weight program? Nor every patient would like complete meal replacement program. 

RESPONSE: Thank you—one of the many challenges in helping people to manage diabetes is inconsistency in available resources. We saw a similar response in the original Diabetes Prevention Program. That being said this could serve as a call for investment in non-pharmacologic programs that if provided more universally could provide personal and public health benefits. This would need to be a system -not a practice response.

As stated throughout the paper, the authors are not suggesting any one treatment option is superior to another or a proper fit for all patients. These limitations have been expanded on in the section titled “Room for Improvement: Limitations and Future Considerations”

  1. instead of using 'target are not met too often', consider using 'after failing to reach targets'

RESPONSE: Thank you. This has been changed.

  1. in IIT, yes, early insulin protects pancreatic b cell from damage, but what about the weight gain with insulin. what is the plan to get people off insulin and make them lose weight? Many patients are not willing to go on insulin. it doesn't mention how long they will be on insulin. this approach of insulin therapy is rarely used these days for diabetes remission, since Bgs are good on insulin. 

RESPONSE: This is an excellent point and one we will clarify. Yes, with typical use insulin is associated with weight gain. However, in studies (including those of the authors) where insulin is used as initial therapy and only for a short time (4-12 weeks), weight gain is not seen. This is the necessity to maximize benefits while minimizing risk.

Many patients are also not willing to proceed with surgery or intensive dietary replacement either. As described in this paragraph, significant differences in anthropometry and weight gain were not found in these studies. The durability of therapy remains a question and this is addressed in the revision to the “Remission with Maintenance Therapy” section.

  1. what is the reference for 80% of b cell function lost. that seems a high number.

RESPONSE: Reference 51 - DeFronzo, R.A., R. Eldor, and M. Abdul-Ghani, Pathophysiologic approach to therapy in patients with newly diagnosed type 2 diabetes. Diabetes Care, 2013.

  1. The sub paragraph ' therapeutic lifestlye change' has been used twice in the article. please consider renaming it. 

RESPONSE: Thank you. This has been changed to “Intensive Lifestyle Modification”

  1. There is talk about triple combo therapy with metformin/pioglit/exenatide, which is almost 10 years older therapy and study. The newer 2nd gen agents like GLP1, GLP-GIP combo, SGLT2, metformin all have shown to have varied degrees of weight loss. There is not much info or studies on these patients. ozempic or mounjaro{step or sustain or surpass trials]  trials showing dm remission have not been mentioned. DM remission paper cannot be complete without mention of these newer gen medications which have changed the landscape of DM therapy. 

RESPONSE: Thank you-this is an important suggestion. You are correct the goal of diabetes remission has waxed and waned over the years as has the focus in clinical trials. This is starting to be a focus for the recent GLP-1Ra/twincretin trials. We were hoping to discuss classes as much as possible rather than individual agents in a class, but we will definitely add the value of the newer medications in making remission a stronger possibility.

Please see the newly added paragraph in the “Combination Therapy” section. These newer gen medications certainly have the potential to change the landscape of how we treat T2DM and address pathophysiology.

  1. Consider incorporating this paper in your article. 

Lingvay I, Sumithran P, Cohen RV, le Roux CW. Obesity management as a primary treatment goal for type 2 diabetes: time to reframe the conversation. Lancet. 2022 Jan 22;399(10322):394-405. doi: 10.1016/S0140-6736(21)01919-X. Epub 2021 Sep 30. Erratum in: Lancet. 2022 Jan 22;399(10322):358. PMID: 34600604. 

RESPONSE: Thank you. This has been incorporated in the future directions section.

  1. This article in its current form seems very superficial, and doesn't dive deep. There are many good aspects of the paper, and the topic chosen is important and relevant. 

RESPONSE: Thank you for your thoughtful review and excellent comments. We agree there is much more to discuss. In this manuscript, we were hoping to sound the alarm and provide enough detail to get people’s attention and will follow with more detail in each arm for future manuscripts so people who want more data to explain more thoroughly.

Hope you can make the paper even better. 

RESPONSE: Thank you!

Round 2

Reviewer 3 Report

This looks great.  

There were lot of changes so it was hard to track all the changes with the red underline, but overall seems great with all the necessary changes incorporated. 

Please proof-read 1 final time before submission and its good to go.